



# The Pollino 2012 seismic sequence: clues from continuous radon monitoring

Antonio Piersanti[1], Valentina Cannelli[1], and Gianfranco Galli[1]

[1]Istituto Nazionale di Geofisica e Vulcanologia, Rome, ITALY

*Correspondence to:* Antonio Piersanti (antonio.piersanti@ingv.it)

**Abstract.** The 2012 Pollino (Calabria, Italy) seismic sequence, culminating in the $M_W$ 5.2 earthquake of October 25, 2012, is investigated exploiting data collected during a long term continuous radon monitoring experiment performed in the epicentral area from late 2011 to the end of 2014. We analyze data collected both using a phenomenological approach based on quantitative evidence and a purely numerical analysis including: *i)* correlation and cross-correlation investigations; *ii)* an original

approach aimed to limit the impact of meteorological parameters variations on the interpretation of measured radon levels; *iii)* a change point analysis; *iv)* the implementation of an original detection algorithm aimed to highlight the connections between radon emission variations and major seismic events occurrence. Results from both approaches suggest that radon monitoring stations can be subject to massive site effects, especially regarding rainfall, making data interpretation harder. The availability of long term continuous measurements is crucial to precisely assess those effects. Nevertheless, statistical analysis shows a

viable approach for quantitatively relating radon emanation variations to seismic energy release. Although much work is still needed to make radon timeseries analysis a robust complement to traditional seismological tools, this work has identified a characteristic variation in radon exhalation during the preparation process of large earthquakes.

## 1  Introduction

One of the most challenging problems in seismology is presently the study of preparatory processes for strong earthquakes.

Seismometric data still represent the most informative observable available to researchers who investigate their association with signals emitted by faults before catastrophic ruptures. In this respect, new features in seismometric records have been discovered and studied in the last years (*Guilhem et al.*, 2008; *Lucente et al.*, 2010; *Fuchs et al.*, 2014). Beside seismometric recordings, slow deformation observations and laboratory experimental simulations contributed to give new important pre-seismic information (*Chlieh et al.*, 2004; *Liu et al.*, 2004; *Tserolas et al.*, 2012; *Jebur et al.*, 2014; *Spagnuolo et al.*, 2015).

Nevertheless, the physical processes taking place on time scales ranging from few years to few hours before the seismic rupture still remain mostly unknown.

Evidence gathered in recent years indicates that, in specific seismotectonic settings, fluid transport and dynamics could play an important role in seismogenic processes (*Miller et al.*, 2004; *Stefansson*, 2011; *Lewicki et al.*, 2014; *Shelly et al.*, 2015). In these seismogenic systems, the study of transient signals associated with fluid migration (markers) becomes particularly

significant. Among all the possible transient signals, the radioactive nature of radon makes it a potentially extremely efficient





marker to study and monitor fluid flows. Indeed, radioactive detectors are generally quite efficient and accurate instruments and their implementation and installation requirements make them also particularly competitive in terms of operating costs. A radon monitoring station equipped with meteorological sensors presently costs almost one order of magnitude less than a $CO_2/O_3$ geochemical station (*Bourcier et al.*, 2011; *Celia et al.*, 2015). The cost factor becomes particularly important considering that

the experience in operating seismometric and geodetic observational networks taught us that, in order to achieve high quality results, instrumentally dense networks are needed.

    From the beginning of 2010 the Pollino Range area, in the southern Apennines on the border between Calabria and Basilicata, has experienced a seismic sequence. The seismic activity alternated frequent periods of intense output with others of relative quiescence and culminated on October 25, 2012 with a $M_W$ 5.2 mainshock (*Tertulliani and Cucci*, 2011; *Totaro et al.*, 2015).

From 2010 to the end of 2014 about 5,000 events (mostly $M_L \leq 3.0$) were recorded (*ISIDe*, 2010). The hypocenters clearly show two main clouds (see Fig. 1). A western cluster which includes most of the seismicity (the $M_W$ 5.2 mainshock too) and seems consistent with a normal faulting trending NNW and dipping WSW. A eastern one, where a $M_W$ 4.3 earthquake occurred on May 28, 2012, does not clearly exhibit instead a definite fault plane (*Totaro et al.*, 2013, 2015). During 2014, two other significant events took place in the area: $M_W$ 3.7 and $M_W$ 4.0 earthquakes on June 4 and 6 on the western and eastern

cluster, respectively.

    In late 2011, we started a long term experiment in the Pollino area of Southern Italy, installing a high sensitivity, high efficiency active radon monitoring station based on a Lucas cell (*Lucas*, 1957; *Semkow et al.*, 1994; *Abbady et al.*, 2004). In November 2012, a second station was installed a few kilometers away from the first one.

    Several world-wide compilations of radon emission anomalies that could be associated with variations in the seismic activity

and/or occurrence of a single earthquake are available in the literature (see *Cicerone et al.* (2009) for a review). In recent years, laboratory experiments gave unambiguous evidence of the relation between the rock state of stress and variations in the radon emanation properties (*Tuccimei et al.*, 2010; *Mollo et al.*, 2011). Nevertheless, highlighting the footprint of internal seismogenic processes in radon timeseries collected in the field is far from being a solved (or even well defined) problem.

    It is widely accepted that meteorological parameters play an important role in modulating soil radon emanations (*Singh et al.*,

1988; *Zmazek et al.*, 2003; *Cannata et al.*, 2009; *Jaishi et al.*, 2014; *Piersanti et al.*, 2015). But, as evidence grows, it becomes clearer that this relation is complex and strongly site dependent, so it cannot be steadily assessed. Even the relative importance among the main relevant variables (temperature, precipitation, pressure) in modulating the radon emissions cannot be univocally determined and it is likely to be site dependent, since different analyses led to different results (i.e., *Zafrir et al.*, 2013; *Jaishi et al.*, 2014; *Kumar et al.*, 2015; *Piersanti et al.*, 2015).

In the following, we propose an articulate approach, taking advantage of different investigative tools, to better assess the questions described above. In particular, we will consider the problem both from a quantitative phenomenological point of view and by means of suitable numerical analyses. The presentation of our results is organized as follows: in Sect. 2 we describe the observational setup, the collected radon timeseries and some phenomenological insights about the impact of meteorological conditions on the detected signal. In Sect. 3 we analyze timeseries by means of different numerical approaches:

namely, in Subsect. 3.1 we perform a correlation and cross-correlation analysis between radon emanation observations and the



other relevant observables (meteorological parameters and seismic moment release) and successively we outline an approach aimed to reduce meteorological effects in the measured radon timeseries; in Subsect. 3.2 we investigate the potential predictive capability of the radon signals, testing the possibility of highlighting in advance the occurrence of the major events of the seismic sequence in the Pollino area from the radon timeseries analysis. Finally, in Sect. 4 we discuss and summarize all our

findings.

## 2   MMN and MMNG sites

We installed two radon monitoring stations in the Pollino area, equipped with prototype detectors based on a Lucas cell, that acquired continuously radon concentration data, with a sampling interval of 2 hours. Station MMN was co-located with the homonymous seismic station belonging to the INSN, Italian National Seismic Network, at Mormanno ($39°53'58.6''N$

$15°59'25.5''E$) in December 2011, at about 921 m above sea level. Station MMNG was installed in October 2012 (just after the $M_W$ 5.2 event) about 3.0 km east of MMN ($39°53'8.1''N$ $16°1'33.6''E$), at about 858 m above sea level. Both stations are shown in Fig. 1 with green triangles. The complete timeseries and technical features characterizing the MMN and MMNG stations are reported in the Supplementary Information appendix.

Station MMN shows a high variability in radon concentration, with sharp peaks and rapidly changing values ranging from a

few tens up to 2500 $Bq\,m^{-3}$ (see Supplementary Information Fig. S1), while MMNG station has lower concentration values (up to 600 $Bq\,m^{-3}$) and a trend ascribable to a major seasonal correlation with temperature (see Supplementary Information Fig. S2), as laboratory tests (*Iskandar et al.*, 2004) and long term radon monitoring studies (*Cannata et al.*, 2009; *Jaishi et al.*, 2014; *Pitari et al.*, 2014; *Piersanti et al.*, 2015) would indicate.

The evidence of the impact of meteorological parameters on radon observations and at the same time the strong site-

dependent nature of the characteristics of radon emissions introduce uncertainties into the comprehension of the problem. These complexities suggest the problem should be approached from a phenomenological point of view in order to supplement the indications retrieved by means of a purely quantitative analysis. First of all, we focus on the "sealing" effect induced by precipitation on soil radon emanation. Such effect has already been suggested and established by several studies (i.e., *Inan et al.*, 2012; *Kumar et al.*, 2015), and its impact in the MMN timeseries seems particularly evident. Figure 2 shows a collection of

selected periods from MMN timeseries (radon in concentration [$Bq\,m^{-3}$]/115 min) corresponding to major rainfall episodes. From Fig. 2 it is clear that, after a major precipitation episode (red ellipses), radon concentrations drastically fall by a factor greater than 10 up to a factor of almost 100. Precipitation, as well as all the meteorological parameters discussed here, is obtained as short term (12-24h) weather forecast by an Italian weather forecasting site (http://www.ilmeteo.it/). Figures 2a, 2b, 2c and 2d represent fall-winter heavy rain events, that are common events in this region (*Federico et al.*, 2008;

*Terranova and Iaquinta*, 2011; *Vennari et al.*, 2014), whilst Figs. 2e and 2f show spring-summer time-windows, when shorter and less intense rain episodes occur. Despite the different magnitude of precipitation episodes, similar reduction effects in radon emission can be seen in fall-winter as well as in spring-summer periods. Moreover, it can be seen that during prolonged dry periods, independently from the season, radon concentration peaks are more pronounced (yellow rectangles). For the MMNG



station the reduction effect of rainfall on radon observations seems less marked, but it is still present. Figures 3a, 3b, 3c and 3d show selected fall-winter and spring-summer periods for MMNG, respectively. In this case, though the reduction of radon emission with precipitation is still present (Fig. 3a), heavy rain events cannot be clearly separated from radon concentration peaks, being sometimes overlaid (yellow rectangles) (Figs. 3b, c and d).

## 3   Analysis of radon timeseries

In the following we try both to assess the impact of meteorological parameters on radon signals on a quantitative basis and to outline an original approach aimed to remove (or at least mitigate) the effects of meteorological events on the detected timeseries. Our goal is to maximize the informative power of radon emanation variations potentially related to a variation in seismic energy release.

Even though the effects of meteorological conditions on temporal radon timeseries have been investigated for the last fifty years by means of different approaches and methodologies (*Singh et al.*, 1988; *Zmazek et al.*, 2003; *Piersanti et al.*, 2015), a clear assessment and a solid interpretation has not been univocally established yet.

For the following analyses, we decided to use only radon timeseries from station MMN, since it was the only one installed before the main events of the sequence ($M_W$ 4.3 on May 2012 and $M_W$ 5.2 on October 2012), corresponding to the major changes in cumulative seismic moment release rate (Fig. 4c). From data collected in the time window from April 2012 to December 2012 (Fig. 5a), that includes the two major seismic events, we note that in correspondence of these two change-points the radon emanation increased a few days before the seismic events. Both the average amplitude and duration of such increases appear to scale with the magnitude of the corresponding earthquakes, as highlighted in the two yellow rectangles of Fig. 5a. The apparent discontinuity in the radon increase just after the $M_W$ 5.2 seismic event is likely to be associated with a major precipitation episode right after the earthquake occurrence. Figures 5b and 5c show in detail the time windows corresponding to the two seismic events.The intensity of radon emanation sharply increases about 24-48 hours before the occurrence of both earthquakes, reaching similar peak values (800-900 $Bq\,m^{-3}$) and then, in the case of the May 28, 2012 $M_W$ 4.3, it returns to previous values after about 7 days, while, after the mainshock of the October 25, 2012 $M_W$ 5.2 event, observed values continue to increase up to about 1600 $Bq\,m^{-3}$ for more than 30 days after the earthquake (except, as described above, for the first days of November, when a major precipitation event flattened down radon levels).

### 3.1   Correlation and cross-correlation analysis

In order to quantitatively assess the phenomenological evidences described above by means of numerically objective procedures, we perform a series of statistical evaluations on our dataset. Figure 4 shows the whole timeseries employed in the statistical analysis filtered with a 14-days moving average. All the moving averages employed in our computations are evaluated backwards (i.e., average at day $d_i$, employs only the previous ($d_i$ −14) days). Figure 4a represents the radon concentration (black line) and rainfall (red line), Fig. 4b shows temperature (black line) and pressure (red line). Figure 4c shows the cumulative seismic moment release (black line) with the seismic moment release (red line).



Since the Pearson coefficient reflects mainly a linear relationship between variables, we estimated the correlation between variables using both the Pearson coefficient (*Hollander et al.*, 2014) and a non-parametric correlation coefficient (*Kendall*, 1970). The two approaches yield virtually identical results, so we show here only the classical Pearson analysis. We performed both a correlation analysis between radon and environmental parameters and a cross-correlation analysis between radon, mete-

orological parameters and seismicity. All analyses look for a linear relationship between two variables but the cross-correlation considers it as a function of the time-offset of one relative to the other. Formally cross-correlation function reads (i.e., *Chatfield*, 2004):

$$CC_{uy}(k) = \begin{cases} \frac{1}{N} \sum\limits_{t=1}^{N-k} (u_t - \overline{u})(y_{t+k} - \overline{y}) & k = 0, 1, \ldots, (N-1) \\ \frac{1}{N} \sum\limits_{t=1-k}^{N} (u_t - \overline{u})(y_{t+k} - \overline{y}) & k = -1, -2, \ldots, -(N-1) \end{cases} \tag{1}$$

where $N$ is the series length, $u_t$ and $y_t$ are the two time series, $\overline{u}$ and $\overline{y}$ are their sample means, and $k$ is the lag. Differently

from Pearson linear correlation, the cross-correlation coefficient is not normalized a-priori: in order to grant compatibility with the previous analyses, we normalized the cross-correlation coefficient here so that it varies between -1 and 1 and set the lag range between -40 and 40 days.

We decided to exclude rainfall from this analysis since, differently from other meteorological variables, it is intrinsically characterized by a strongly discontinuous, spike-like behavior being the majority of the sampling times characterized by a null

value. In fact, during the time window of our most relevant analyses, we have null rain values ranging from 65 % to 75 % of the sampling intervals (to compare for instance with less than 10 % of days with null seismic moment release). This makes correlation and cross-correlation analysis inadequate approaches to evaluate the relationship between radon concentration and rainfall.

The results regarding the correlation analysis in terms of Pearson coefficient are summarized in Table 1. For all considered

cases, we report both global cumulative value ($G$) corresponding to the entire acquisition window (2012-2014) and separate results for each year (2012-2013-2014). The Pearson coefficient $\rho$ shows a significant level of negative correlation only between radon concentration and temperature with values ranging from $-0.6$ to $-0.2$. The value of the Pearson coefficient for pressure, even though coherent both in sign and in magnitude for each time window, is nevertheless statistically compatible with zero.

Within the cross-correlation analysis, whose results are shown in Fig. 6, we include also the seismic moment release $M_0$,

since for this physical variable a lagged approach is able to consider also a causal relationship in addition to an instantaneous feedback among variables (*Box and Jenkins*, 1976; *Piersanti et al.*, 2015). Figure 6 is arranged in nine panels: from left to right the cross-correlation between radon and temperature, pressure and seismic moment release, respectively, are presented, while the rows represent 2012, 2013 and 2014 time-windows. No sharp and isolated peak is observed in Fig. 6, indicating that no clear cross-correlation scenario can be deduced from this analysis. Nevertheless, we can confirm the correlation pattern

described above: the cross-correlation function between radon concentration and temperature does not show clear preferences for a lag time, but it is almost always characterized by negative values, while the cross-correlation between radon and pressure timeseries varies in time, with value always below the 99 % confidence level. The confidence level is defined as the value of




the Pearson coefficient $\rho$ for which the probability of obtaining a cross-correlation greater than or equal to $\rho$ for uncorrelated data is equal to 1 (*Chatfield*, 2004) and is represented in Fig. 6 by the grey lines. The cross-correlation function between radon concentration and seismic activity shows a significative positive peak during 2012 (when the major seismic events occurred), with a maximum value of 0.5 in correspondence of a 21 days delay forward of radon concentration (Fig. 6, panel upper right).

Of course the relationship between variations of radon emanation and seismotectonic processes would be better assessed if we would be able to remove, or at least reduce, the bias of meteorological parameters on the radon measured concentration. To this aim, we implement an empirical correction procedure for temperature, pressure and precipitation variations. Basically, given an observed radon concentration value $Rn_{obs}$, taken at time $t$ when a temperature $T$, an atmospheric pressure $P$ and a precipitation level $R$ have been registered, we define a corresponding meteorological-corrected concentration $Rn_{cor}$ as:

$$Rn_{cor} = Rn_{obs} \times C_P \times C_R \times C_T$$

where $C_P$, $C_R$ and $C_T$ are positive correction factors obtained as a simple linear interpolation from the minimum detected values of T, P and R in a selected time window where $\{C_P = C_R = C_T = 1\}$ (that is to say there is no-correction), to the maximum detected values in the selected time window where $\{C_P = C_{P_{max}}; C_R = C_{R_{max}}; C_T = C_{T_{max}}\}$. The optimal value of $C_{P_{max}}$, $C_{R_{max}}$ and $C_{T_{max}}$ can be obtained by maximizing the cross-correlation function for the selected time window (of course a time window including a significant seismic activity must be selected). We want to note that the subscript $max$ above stands for maximum magnitude of the correction, not for maximum absolute value of the correction parameter $C_i$. Indeed, if the correction factor corresponding to the maximum value of a given meteorological parameter $C_i$ is $> 1$, it means a negative correlation between radon and that parameter, the opposite if the correction factor $C_i$ is $< 1$. Since it is reasonable to consider the possible connection between radon concentration variations and seismotectonic processes as dependent from the seismic source-observer distance (*Dobrovolsky et al.*, 1979), we have implemented in the correction procedure also the possibility of

weighting for the epicentral distance (*Hauksson and Goddard*, 1981; *Einarsson et al.*, 2008). Again, given an earthquake with seismic moment $M_{0_{obs}}$ occurred to an epicentral distance $r$ from station MMN, we consider a corresponding distance-weighted value $M_{0_{wgt}}$:

$$M_{0_{wgt}} = \frac{M_{0_{obs}}}{r^w}$$

where $w$ is a positive weighting factor ($w$=0 means no correction for epicentral distance).

In Fig. 7 we show the effects of our correction procedure on the cross-correlation function. The extrapolation of the optimal

values for the correction parameters $C_{P_{max}}$, $C_{R_{max}}$, $C_{T_{max}}$ and $w$ was performed by means of the `MINUIT` package (*James*, 1998), which implements a variable-metric method with an inexact line search, a stable metric updating scheme and a positive-definiteness check (*Fletcher*, 1970). The search domain for $C_i$ and $w$ was limited in the range between 0.1 to 10 to avoid unphysical solutions. This procedure has been applied with two different time windows, both including the two main events and the active part of the sequence (May 2012 $M_W$ 4.3 and October 2012 $M_W$ 5.2): the first time window (tw-1) covers a

whole year from January 2012 to January 2013, while the second (tw-2) focuses on the most active part of the seismic sequence





from April 2012 to January 2013. As can be seen from Fig. 7, the proposed correction procedure significantly increases cross-correlation peaks for both time windows (indicated as tw-1 corrected and tw-2 corrected). Notably, the increase is greater for the larger time window where a lower (but still significant) peak cross-correlation value was obtained, while the time lag of the peak remains completely unchanged after the correction, indicating that the variation of radon intensity seems to follow the

variation in seismic moment release. In Table 2 the correction coefficient values maximizing the cross-correlation peak in the two time windows tw-1 and tw-2 are reported. From the tabulated values we note that: *i)* the correction values for the rainfall lie in both cases at the top of the searching domain ($C_{R_{max}}$=10 for tw-1 and $C_{R_{max}}$=9 for tw-2), i.e., rainfall is strongly anti-correlated with radon emanation, confirming the phenomenological analysis in previous Sect. 2; *ii)* the correction values for the temperature are always greater than 1, confirming that for MMN station temperature is anti-correlated with radon emanation

(see above in this same section); *iii)* the correction values for the pressure oscillate about $C_{P_{max}}$=1, confirming the lack of a clear correlation regime between pressure and radon emanation for this station.

### 3.2   Change point analysis and detection algorithm

The problem of detecting changes in timeseries is well known in climate literature: the definition and identification of discontinuous steps, or change points, may be subjective and it also depends on the form of the trend one expects between changes.

Several methods have been implemented to solve the change point problem both for short and long climatic timeseries. We refer the readers to *Reeves et al.* (2007), in which the literature about the change points methods is widely reviewed and discussed.

We applied to the measured radon intensity timeseries an algorithm developed in the realm of Earth's climate system studies in order to calculate, by means of a bayesian approach, the posterior probability of multiple change points in a generic climatic timeseries (Bayesian Change Point algorithm, (*Ruggieri*, 2013), BCP hereinafter). Once the algorithm has identified

an arbitrary number of change points in our timeseries, whose maximum is an input parameter of the algorithm ($k_{max} = 6$ in the following) , our primary interest is to verify if the detected change points in the radon timeseries are consistent with corresponding changes in cumulative seismic moment release rate (*i.e.*, major earthquakes).

Applying the BCP algorithm to the whole MMN timeseries, we obtain an indication of most likely two change points that are potentially associable with the two largest events of the sequence. Figure 8 show the 14-days moving averaged timeseries of

radon intensity (solid black line) along with the change point regression model (dashed green line); the locations of the change points are displayed as red spikes, whilst earthquakes occurrences are displayed as yellow stars. The algorithm has furthermore the ability to provide an uncertainty estimate in locating a change point: in this case the height of the two considered spikes (the second and the third in Fig. 8) indicates a probability equal to 0.33 for the change point corresponding to the the $M_W$ 4.3 on May 2012 and a probability equal to 0.57 for the change point corresponding to the $M_W$ 5.2 on October 2012. The second

change point occurs on May 8, 2012, 20 days before the $M_W$ 4.3 May 28, while the third change point occurs on October 22, 2012, 3 days before the $M_W$ 5.2 October 25 event.

The different time advances of the change points found by the BCP algorithm with respect to the two associated earthquakes occur (20 days and 3 days before, respectively) is not determinant for our investigations, since the dynamics of radon emission is intrinsically complex, as shown also by *Jaishi et al.* (2014); *Kumar et al.* (2015); *Piersanti et al.* (2015). Neverless, it could





be useful to get further insight into the relationship between radon and seismicity, employing the same BCP algorithm on the the cumulative seismic moment release timeseries, in order to check the possibility of finding significant variations in seismic moment release other from the trivial ones (*i.e.* coincident with a major seismic event). The result is shown in Fig. 9: in this case the rates changes, that are clearly visible *a priori*, are all found by the algorithm with a probability near to 1. While the

$2^{nd}$ and the $4^{th}$ change points clearly identify the two earthquakes, the $1^{st}$ and the $3^{rd}$ change points seem, instead, to identify the beginning of a preparatory phase of the two events. The first occurs on February 20, 2012 ($1^{st}$ red spike in Fig. 9) and the third on August 18, 2012 ($3^{st}$ red spike in Fig. 9). We note that the temporal difference (about 70 days) between each of these two change points and the change points estimated by the BCP algorithm for the MMN timeseries (the two blue dashed vertical lines in Fig. 9) is comparable. In this respect, radon concentration variations could be sensitive to the internal processes taking

place during the preparatory phase of an earthquake.

     We point out the fact that a standard change point analysis uses always the whole timeseries, since to identify a change point at a time $t_i$ the algorithm processes also data at $t > t_i$. This is a limitation because the algorithm cannot be employed for predictive purposes. To overcome these limitations and most of all to extend the range of our investigations, we implemented an original detection algorithm that potentially could be used in real time analyses. A schematic flow chart of the algorithm is

shown in Fig. 10. It basically works on a simple two stage condition: *i)* the radon daily average (DA) exceeding by a factor ($p_1$) the two-weeks moving average (MA) and *ii)* the moving average (MA) successively increasing by a factor ($p_3$) for a given time-window ($p_2$). When both conditions are satisfied, an alarm is issued at day (i+$p_2$) (red font in box of Fig. 10). If an $M > 4.0$ earthquake occurs during 40 days after the alarm have been issued, all the thresholds to issue subsequent alarms are increased by a factor ($p_4$) during a time window proportional to the energy released by the event ($p_5{}^{M_{EQ}}$). The algorithm works only

with five free parameters and there is no limitation to the number of alarms that could be issued and to the time when they could be issued.

     Figure 11 shows the output of our detection algorithm running on the whole MMN timeseries. Issued alarms are represented by red triangles, while yellow stars mark the largest seismic events that occurred in the 40 days following the alarm. For each year, the two greatest seismic events have been also displayed (white stars), regardless of the issuing of an alarm. Incidentally,

except for 2014, in 2012 and 2013 the two greatest seismic events are just the seismic events that occurred in the 40 days following an alarm. Some main observations can be pointed out here: *i)* the algorithm succeeds in forecasting the $M_W$ 5.2 mainshock of October 2012; *ii)* it succeeds in forecasting the two main events of the whole sequence (the $M_W$ 5.2 of October 2012 and the $M_W$ 4.3 of May 2012 that started the most active part of the sequence); *iii)* it succeeds in forecasting the major events for 2012 and 2013, while it fails for 2014; *iv)* it issues only one false alarm in three years. We note also that the time

advance of the alarms with respect to the earthquake occurrence for the two main events of the sequence is remarkably similar to that observed by means of change point analysis.

     Therefore, both the cross-correlation analysis and the change point analysis, as well as the application of our detection algorithm, indicate that a physical relation between the variation of soil radon emanation and seismic moment release exists.While change point and detection algorithm both succeed in finding some useful radon signal before the variation in seismic moment

release, the cross-correlation investigations seem to behold the radon signature after the seismic moment release variation.




Relying on the change point analysis and detection algorithm, we have verified if also the cross-correlation analysis is compatible with a radon signal preceding the seismic moment release signal. To investigate this possibility, we have repeated the procedure described in Subsect. 3.1. In this case we limit the search domain to positive lag values (i.e. radon signal preceding moment release signal), in order to verify if a suitable solution can be found also in this case. As Fig. 12 highlights, such a
solution exists and, comparing Figs. 7 and 12, it is evident that it is only marginally less significant with respect to the best one. Remarkably, as a confirmation of the previous findings, the correction coefficients associated with this solution (see Table 3) are consistent with these found in Subsect. 3.1. They indicate for radon observations at MMN station a strong anti-correlation with respect to precipitation ($C_{R_{max}}$=9.3 for tw-1 and $C_{R_{max}}$=10.0 for tw-2), a clear anti-correlation with temperature and the lack of a clear correlation with respect to pressure variations.

## 4   Conclusive remarks

We have performed a detailed analysis of the temporal variations of radon emanations from late 2011 to 2014 in a seismically active area during a seismic sequence that culminated at the end of 2012 with a $M_W$ 5.2 event. We exploited several different approaches to carry out our investigations. Namely: *i)* phenomenological analysis; *ii)* correlation and cross-correlation investigations; *iii)* empirical correction of the meteorological parameters effect on radon timeseries and its impact on cross-
correlation; *iv)* change point analysis; *v)* detection algorithm.

We can split the main results of our work in two classes: *a)* those concerning the impact of meteorological parameters variation on the observed radon timeseries and *b)* those concerning the existence of a physical connection between the observed radon timeseries and the seismic moment release temporal variations. Converging indications coming from both classes represent an important outcome of our work. Regarding class *a)*, we have indications that, in the investigated setting, soil radon
emanation is strongly anti-correlated with precipitation and weakly anti-correlated with temperature, while we do not get significant and univocal evidence of correlation (positive or negative) with pressure variations. In this context, approaches *i)*, *ii)* and *iii)* give remarkably consistent indications and we see as particularly significant the agreement between the strength of the correlation evidenced by *i)* and *ii)* and the magnitude of the corresponding correction factor found with *iii)*. These results, when compared with previous findings, confirm that the environmental impact on radon observations is strongly site depen-
dent. The correlation between radon variations and temperature is, in this sense, a clear example: many works found it positive, as several others (including ours), negative. This observation suggests that a specific characterization is needed for each station, when implementing an observational network (see, for example, the dependence on the varying soil characteristics as porosity, permeability, and pre-rain moisture state). Regarding class *b)*, all our analyses univocally indicate the existence of a non-accidental correlation between the temporal evolution of soil radon emanation and seismic moment release. The primary
output of approach *ii)* suggests that the radon signal follows the seismic moment variation, while approaches *i)*, *iv)* and *v)* indicate that it is possible to retrieve the radon signal also before the seismic moment variation. Remarkably, we have found that even if approach *ii)* gives as primary result a shifted forward temporal correlation, nevertheless, also the solution with the radon signal preceding the seismic moment variation is acceptable at a barely lower significance level.



*Author contributions.* TEXT

*Acknowledgements.* TEXT



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





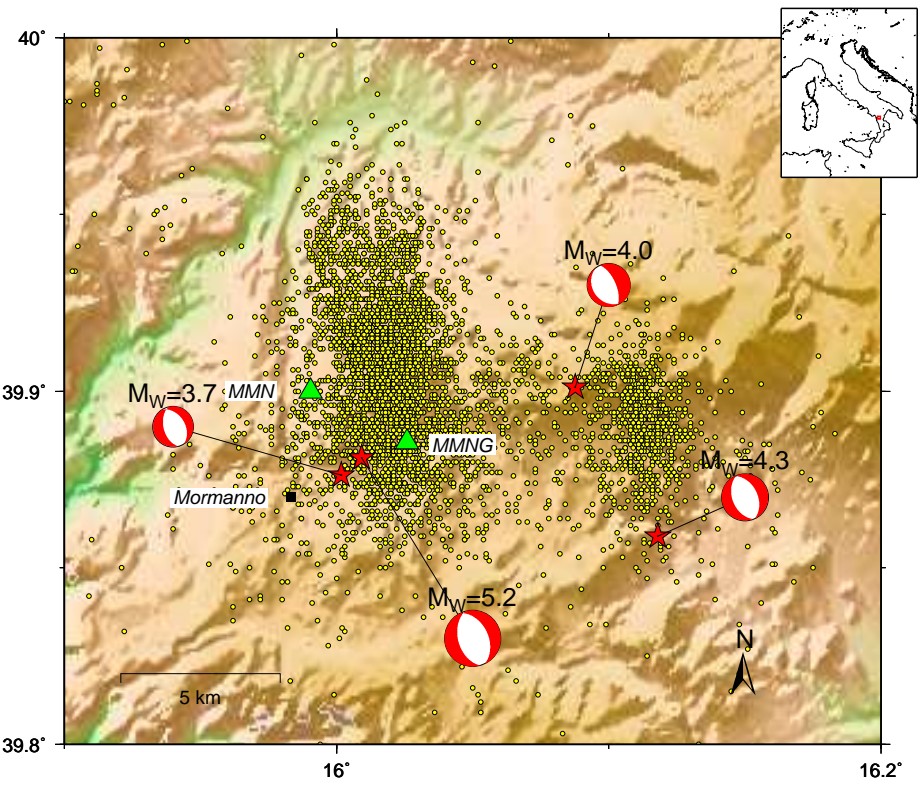

**Figure 1.** Geographical setting of the study area located in the Calabria peninsula, southern Italy (see inset). Green triangles show the location of a radon monitoring stations MMN and MMNG. Yellow circles represent the earthquakes recorded by *ISIDe* (2010) between December 2011 and October 2014 with epicentral distance from MMN (the older station) less than or equal to 15 km (4,800 events). Focal mechanisms of the $M_W$ 4.3 May 28, 2012, $M_W$ 5.2 October 25, 2012, $M_W$ 3.7 June 4, 2014 and $M_W$ 4.0 June 6, 2014 earthquakes (http://cnt.rm.ingv.it/tdmt.html) are also represented.





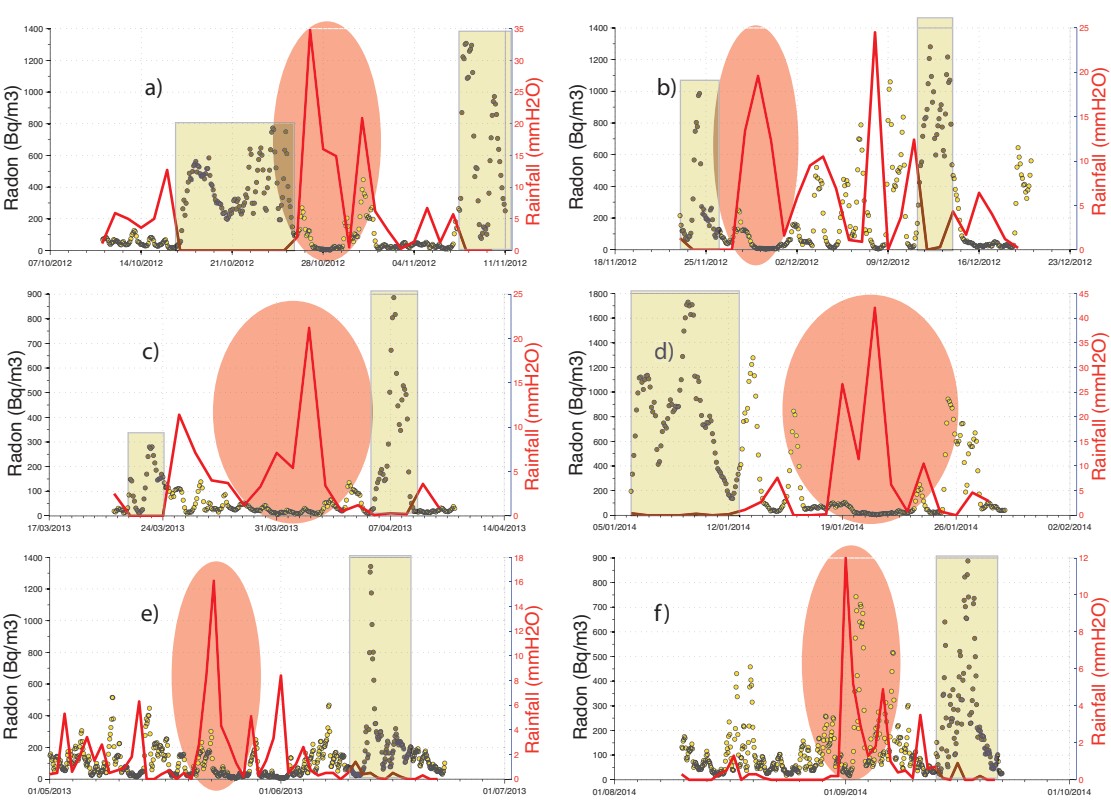

**Figure 2.** MMN radon concentration in $[Bq\,m^{-3}]$/115 min (yellow dots) and daily average rainfall (red line) for some significant fall-winter
(a),b),c) and d)) and spring-summer (e) and f)) periods (see text for details). Red ellipses mark heavy rain events, whilst yellow rectangles
represent radon concentration peaks.





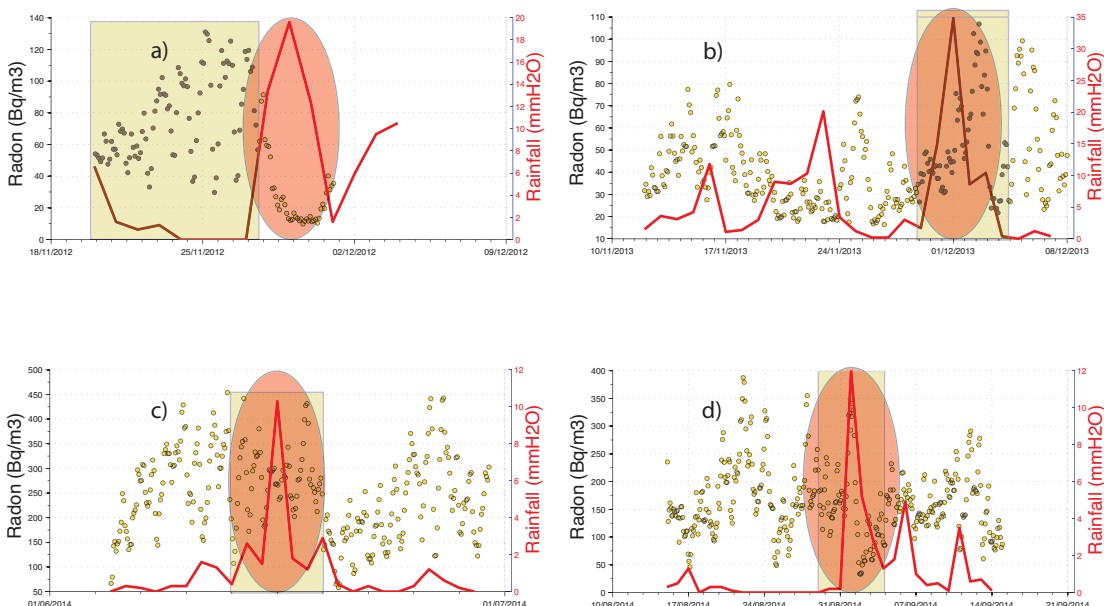

**Figure 3.** MMNG radon concentration in $[Bq\,m^{-3}]$/115 min (yellow dots) and daily average rainfall (red line) for some significant fall-winter (a) and b)) and spring-summer (c) and d)) periods (see text for details). Red ellipses mark heavy rain events, whilst yellow rectangles represent radon concentration peaks.



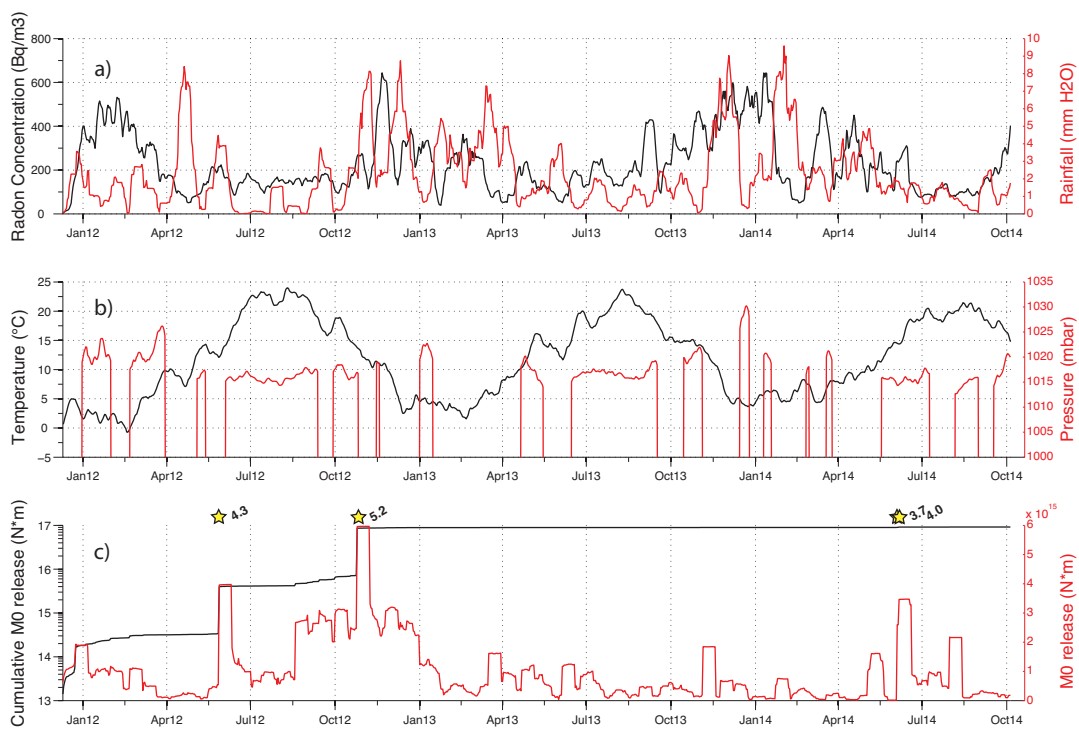

**Figure 4.** a): 14-days moving averaged timeseries of radon concentration at MMN (black line) and of rainfall (red line). b): 14-days moving averaged timeseries of temperature (black line) and pressure (red line). c): 14-days moving average of cumulative seismic moment release (black line, in logarithmic scale) and of seismic moment release (red line). Yellow stars represent the occurrences of the main earthquakes of the sequence.




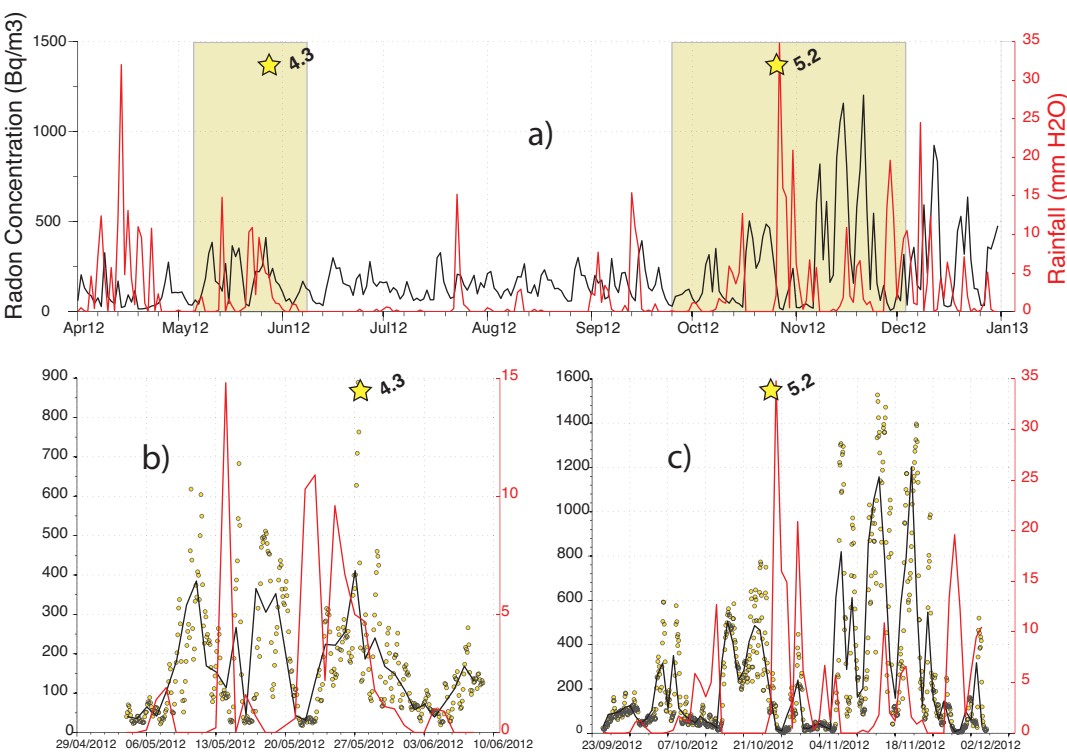

**Figure 5.** a): Timeseries of daily moving averaged radon concentration at MMN (black line) and of rainfall (red line) for the period between April 1, 2012 and December 31, 2012. Yellow stars represent the occurrences of the two main earthquakes of the sequence. b): an enlarged view of the first yellow rectangle of a), from May to June 2012. Yellow dots represent radon concentration at MMN in $[Bq\,m^{-3}]$/115 min, while a black line shows the daily averaged concentration. Daily averaged rainfall levels are represented by a red line. c): An enlarged view of the second yellow rectangle of a), from September to December 2012, as in panel b).





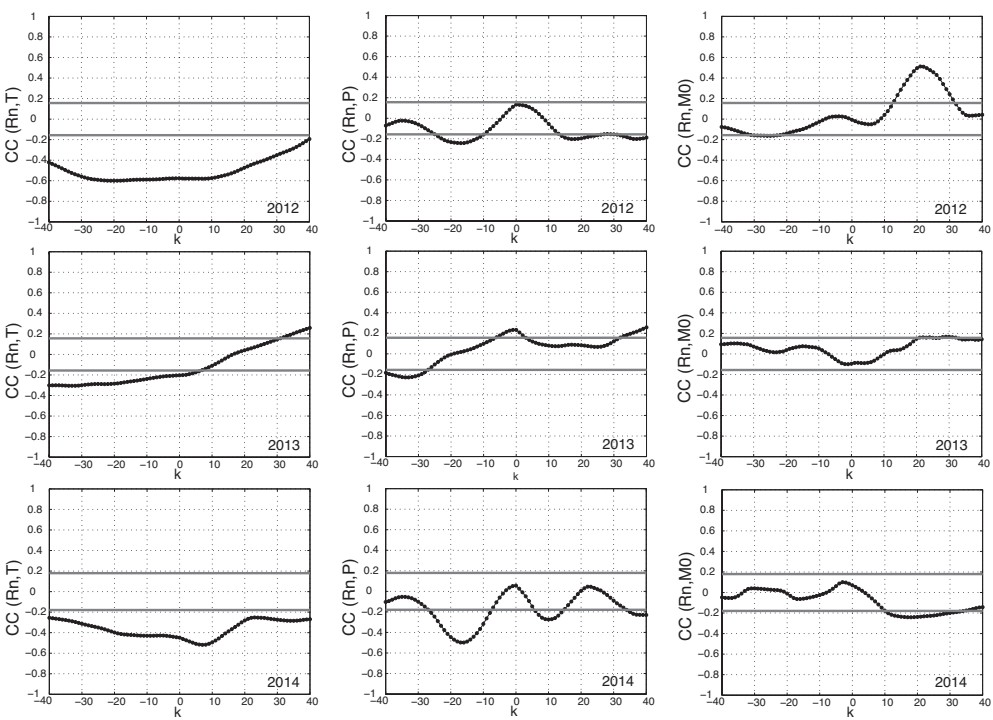

**Figure 6.** Cross-correlation function (CC) evaluated for 2012-2013-2014 separately, between radon concentration Rn and temperature T, pressure P and seismic moment release $M_0$. The CC is evaluated between 14-days moving average filtered timeseries. Horizontal gray lines represent 99 % confidence threshold.





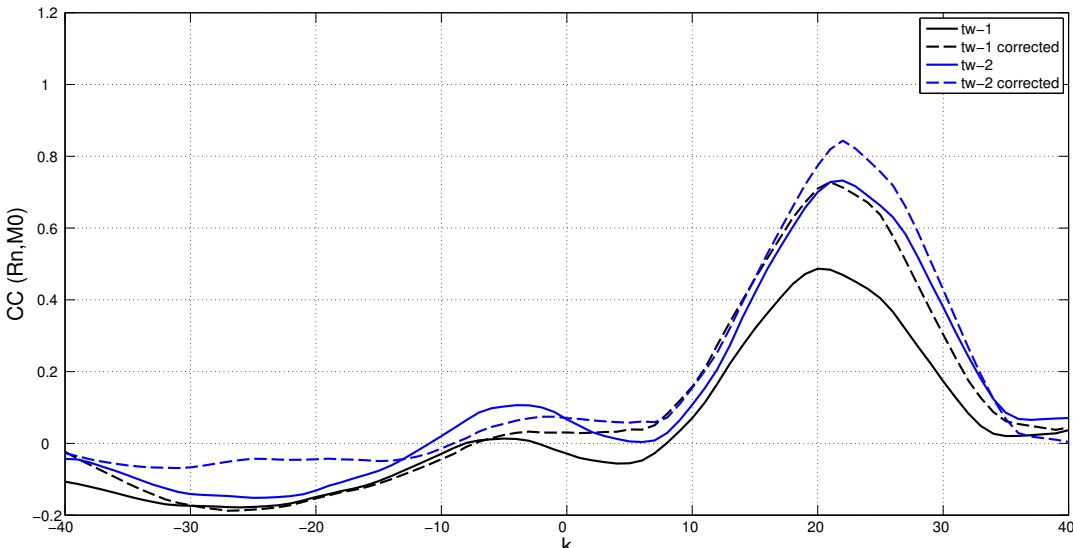

**Figure 7.** Cross-correlation function (CC) between radon concentration Rn and seismic moment release $M_0$ timeseries, filtered with a 14-days moving average and evaluated in two different time windows (black line for tw-1 and blue line for tw-2), with (dashed lines) and without (solid lines) correction coefficients. Both the values of correction coefficients and time windows bounds are summarized in Table 2.

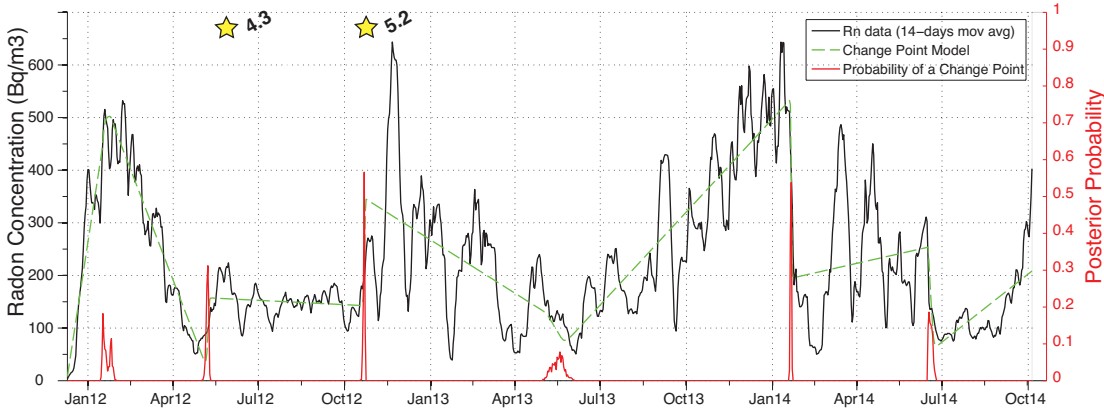

**Figure 8.** Change point analysis applied to timeseries of radon concentration at MMN. The black solid line represents the radon concentration at MMN filtered with a 14-days moving average, while the green dashed line represents the model predicted by the Bayesian Change Point (BCP) algorithm. The red line represents the probability of a change point at each time. Yellow stars represent the occurrence of the earthquakes $M_W$ 4.3 on May 28, 2012 and $M_W$ 5.2 on October 25, 2012.





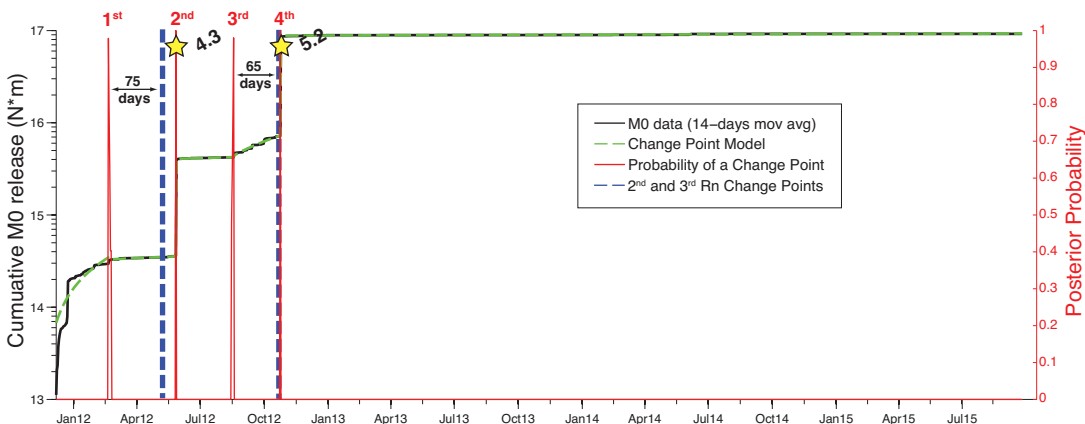

**Figure 9.** Change point analysis applied to cumulative seismic moment release. The black solid line represents the cumulative seismic moment release filtered with a 14-days moving average, while the green dashed line represents the model predicted by the BCP algorithm. The red curve indicates the probability of a change point at each time. The two blue dashed vertical lines mark the occurrence of the second and of the third change point represented in Fig. 8. Yellow stars represent the occurrence of the earthquakes $M_W$ 4.3 on May 28, 2012 and $M_W$ 5.2 on October 25, 2012.




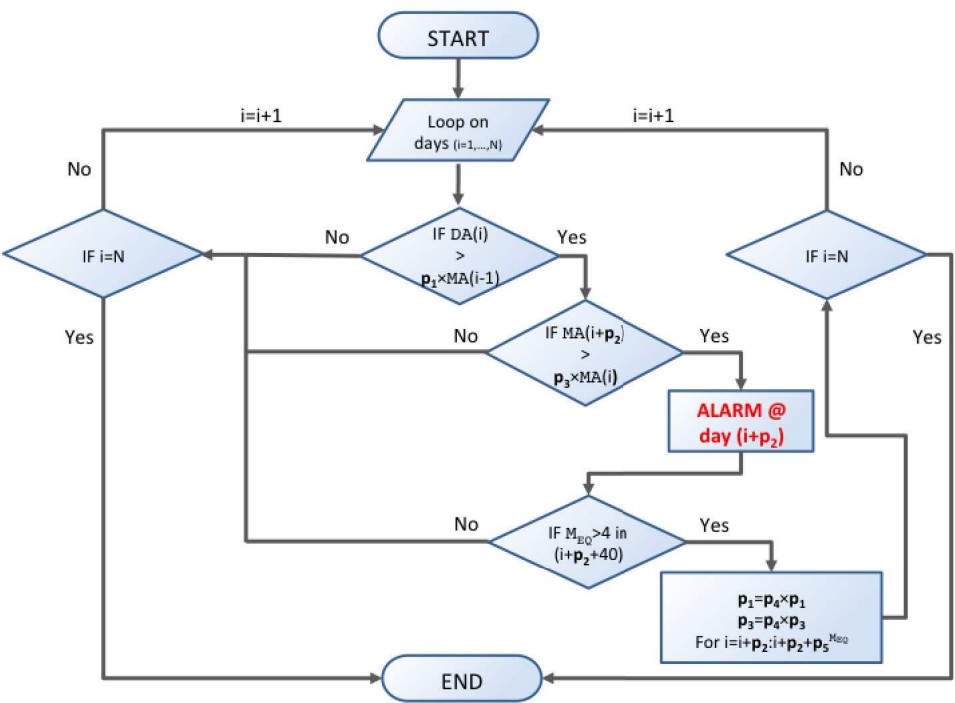

**Figure 10.** Flow chart representing the detection algorithm. $(p_1, p_2, p_3, p_4, p_5)$ are the free five parameters described in the text. DA and MA are the daily and the two-weeks moving average of radon timeseries, respectively. $M_{EQ}$ is the magnitude ($>4$) of the earthquake occurring (if any), during 40 days after the alarm.

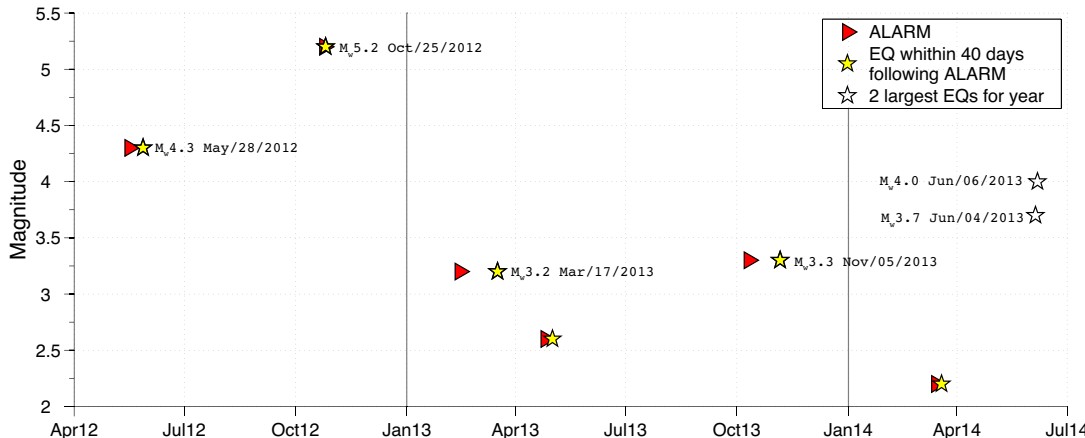

**Figure 11.** Output of the detection algorithm applied to timeseries of radon concentration at MMN. The red triangles represent all the issued alarms, yellow stars represent the greatest seismic events occurred in the 40 days following each alarm. For each year the two greatest seismic events have been also displayed (white stars), with corresponding occurrence date and magnitude.




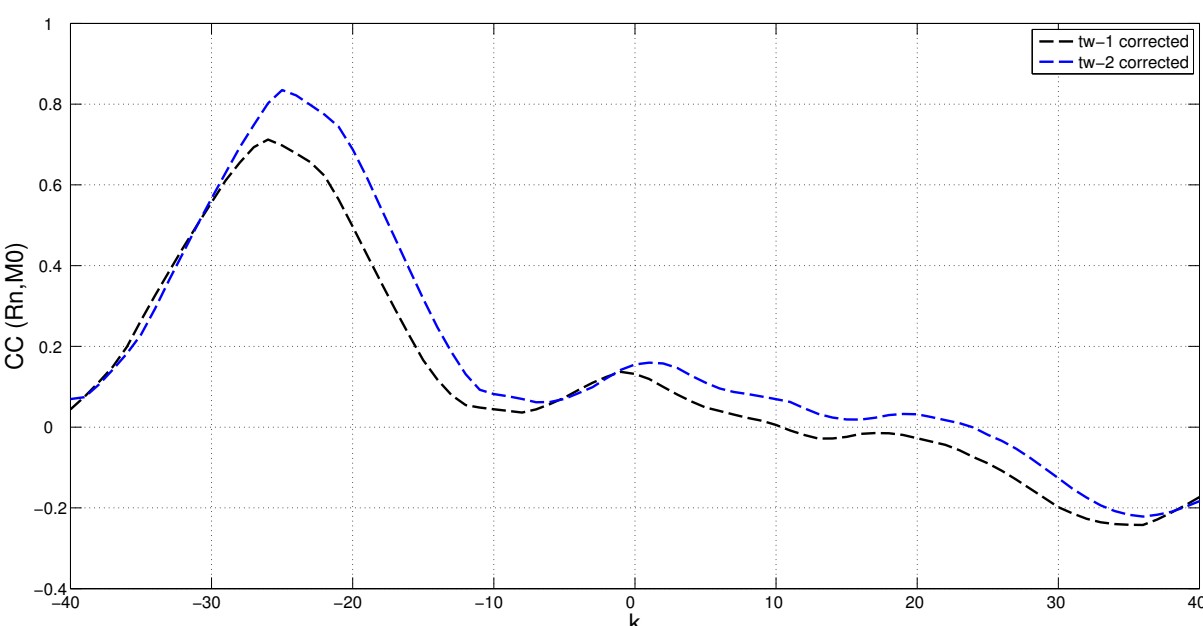

**Figure 12.** The same as Fig. 7, but limiting the search domain of MINUIT only to positive lag values (k) (the corresponding correction coefficients are reported in Table 3).





**Table 1.** Pearson correlation coefficient ($\rho$) between radon concentration timeseries (Rn) and temperature (T), pressure (P) timeseries, evaluated both as global value (G) for the entire acquisition window and as annual value for 2012, 2013, 2014 separately. Rn concentration, T and P timeseries are filtered with a 14-days moving average.

| $\rho$ | G | 2012 | 2013 | 2014 |
|---|---|---|---|---|
| (Rn,T) | -0.46 | -0.58 | -0.20 | -0.46 |
| (Rn,P) | 0.15 | 0.13 | 0.24 | 0.04 |

**Table 2.** Correction coefficients for temperature ($C_{T_{max}}$), pressure ($C_{P_{max}}$), rainfall ($C_{R_{max}}$) and epicentral distance ($w$) maximizing the cross-correlation function (CC) in time-windows tw-1 (Jan2012-Jan2013) and tw-2 (Apr2012-Jan2013).

| | $C_{T_{max}}$ | $C_{P_{max}}$ | $C_{R_{max}}$ | $w$ |
|---|---|---|---|---|
| tw-1 | 2.4 | 4.4 | 10.0 | 1.3 |
| tw-2 | 5.6 | 0.9 | 9.0 | 0.0 |

**Table 3.** The same as Table 2, but limiting the search domain of `MINUIT` only to positive lag values (k).

| | $C_{T_{max}}$ | $C_{P_{max}}$ | $C_{R_{max}}$ | $w$ |
|---|---|---|---|---|
| tw-1 | 5.6 | 0.9 | 9.3 | 3.0 |
| tw-2 | 2.7 | 1.6 | 10.0 | 3.0 |