# Peer review of "Supporting Information for "The Pollino 2012 seismic sequence: clues from long term continuous radon monitoring""

_Solid Earth, 2016_

## Referee Comment (RC1) · R. C. Tiwari (Referee) · 13 Jun 2016

General Remarks:
This research paper reports a significant contribution towards the area of seismic precursory studies, which is one of the most complex phenomenon and a challenge to Science in general and to Geoscience, in particular. One of the reasons being, the site dependent behavior of Geo-chemical parameters. This study assumes significance in view of the fact that clear experimental evidence of relation between the rock state of stress and variations in radon emanation behavior have been reported in the literature, however, it lacks to provide a clear cut understanding about inside soil behavior of radon.

Reviewer Comments:

[Figure]

Comment 1:
This paper is well organized and presents in-depth analysis of 2 selected seismic events out of about 5000 events. The analysis of the experimental data has been done using two approaches namely phenomenological and a purely numerical analysis. The authors have successfully arrived at same or similar conclusions, however, micro-seismic events of magnitude less than 3 and radon emanation have not been considered. It cannot be ruled out that radon anomalies for such events may be indicators of impending massive earthquakes.

Comment 2:
Authors have considered daily average and 14 days moving average data to analyze the radon behavior for 2 stations by different methods, indicates the efforts that have been put by authors to generate the data to understand the phenomenon. It is interesting to see a non accidental correlation between Radon concentrations and seismic moment release. I wish to appreciate the authors for their commendable job.

Comment 3:
In general, the radon peaks showing variations in concentrations for 2 SD (standard deviation) or more that 2 SD are considered to be anomaly due to seismic events, provided that the data is rectified for the false anomaly that may be due to influence of meteorological parameters (Pressure, Temperature, Rainfall etc). I do hope authors have taken care of this aspect. Radon anomalies of both types (rise/fall in radon concentration) are equally important.

Comment 4: I am in full agreement with the authors that a long term data recording/analysis is necessary to be able to arrive at definite trends of behavior of geo-physical/geo-chemical parameters and conclusions that may be worthy to be applied to develop seismic prediction models in future.

Comment 5:
Western side of study area has recorded more number of seismic events and of higher magnitude as compared to eastern side. Explain reasons?

Comment 6:
Whether the meteorological observatory is located on site, or far from the radon monitoring site, if so how far from the site? Please clarify the situation for both monitors.

Comment 7:
Authors are requested to explain reasons for the following:
a) During summer season, Radon concentration is more pronounced. Why?
b) Radon concentration decreases with increase in rainfall. Why?
c) Can you elaborate further on your claim with explanation that 'the discontinuity in the radon increase is likely to be associated with a major rainfall that occurred just after the 5.2 magnitude earthquake'?

Comment 8:
It is very interesting observation recorded by authors that:
For the seismic event of magnitude 4.3, radon concentration gets restored to normal value after 7 days of the event.
For the seismic event of magnitude 5.2, radon concentrations continue to increase for more than 30 days after the event.
Can you explain the possible reasons?

Comment 9:
Typo-graphical suggestions:
Page 1, Line-15: observable may be replaced by observations
Page 1, Line-17: last years may be replaced by recent past

Page 4, Line-25: few may be added between first and days
As far as possible, the curves in Figs. 2,3,4,5 and 8 may be bolded for better clarity in print copy.

Comment 10:
I strongly recommend the publication of this paper in the journal Solid Earth, subject to minor modifications/clarifications as described above, to be incorporated before re-submission.

---

## Referee Comment (RC2) · P. Einarsson (Referee) · 22 Jun 2016

Comments by Páll Einarsson palli@hi.is June 22, 2016

General Remarks: The paper presents results from a radon monitoring project in Italy, a timeseries of radon activity before, during and after a nearby earthquake sequence. Several methods are used to analyse the data and quantify the relationship between radon activity, environmental parameters, and seismicity. This is an unusually thorough analysis and appears to be based on solid methodology. The conclusions appear to be sound, they are carefully stated and supported by the data analysis. In my opinion the paper represents a significant contribution to the studies of earthquake source processes and the failure preparation phase. It should be of considerable interest to the readership of Solid Earth.

[Figure]

I reviewed an earlier version of this paper. The authors have already responded to my earlier remarks in a satisfactory manner.

Reviewer Comments: 1. More technical information is needed. I cannot find any information about the sampling technique. There is no mention of what the carrier fluid is, atmospheric air, ground water, geothermal fluid, geothermal gas? What is the depth of sampling, surface, soil, drill hole? This is particularly important in the light of the strong correlation of radon concentration with meteorological factors, (see e.g. page 6, lines 5-6). This correlation is understandable if the sampling takes place at the surface. It may be possible to reduce the disturbing influence of weather by taking samples from deep drill holes (Hauksson and Goddard, 1981, Einarsson et al., 2008). (I note the words "soil radon emanation" appear on page 8, line 33, and page 9, line 19-20 and 29).

2.I recommend publication of this manuscript, but after minor revisions.

Specific comments: 1. Strange sentence: Page 2, lines 12-13. 2. Strange sentence: Page 2, lines 22-23. 3. Page 8, line 3: ... other than the trivial one ... 4. Page 9, line 32: Complicated sentence. Delete "nevertheless".

Please also note the supplement to this comment:
http://www.solid-earth-discuss.net/se-2016-72/se-2016-72-RC2-supplement.pdf

---

## Author Comment (AC1) · 5 Jul 2016

R. C. Tiwari (Referee)

COMMENT 1: This paper is well organized and presents in-depth analysis of 2 selected seismic events out of about 5000 events. The analysis of the experimental data has been done using two approaches namely phenomenological and a purely numerical analysis. The authors have successfully arrived at same or similar conclusions, however, micro-seismic events of magnitude less than 3 and radon emanation have not been considered. It cannot be ruled out that radon anomalies for such events may be indicators of impending massive earthquakes.

REPLY 1: We confirm that all our analyses involving seismicity (cross-correlation,

change point analysis and detection algorithm) employ ALL events recorded by ISIDe between December 2011 and October 2014 with epicentral distance from MMN less than or equal to 15 km (4,800 events), without any type of selection on magnitude value. Among other things, micro-seismic events (Mw<=3.0) represent 99% of those selected in this work.

COMMENT 2: Authors have considered daily average and 14 days moving average data to analyze the radon behaviour for 2 stations by different methods, indicates the efforts that have been put by authors to generate the data to understand the phenomenon. It is interesting to see a non accidental correlation between Radon concentrations and seismic moment release. I wish to appreciate the authors for their commendable job.

REPLY 2 We sincerely thank Dr. Tiwari for his appreciation

COMMENT 3: In general, the radon peaks showing variations in concentration for 2 SD (Standard Deviation) or more that 2 SD are considered to be anomaly due to seismic events, provided that the data is rectified for the false anomaly that may be due to influence of meteorological parameters (Pressure, Temperature, Rainfall etc). I do hope authors have taken care of this aspect. Radon anomalies of both types (rise/fall in radon concentration) are equally important.

REPLY 3: In the present work we tried to extend the standard "anomaly" analysis by means of a twofold investigative approach: a global correlation and cross correlation analysis and the detection algorithm. The former tries and assesses a not accidental relationship between radon emanation and seismic moment release time series as a whole, while the latter tries and overcomes the usual anomaly notion by means of a more complex and (hopefully!) powerful approach (see also figure 1R). The sense of empirical correction developed in this work is just to take into account (removal or at least reduction) of the bias of meteorological parameters (temperature, pressure and precipitation variations) on the radon measured concentration. The impact that

this correction has also on the cross-correlation analysis represents a good check of its usefulness for us. Presently our detection algorithm is structured only to issue an alarm for a positive variation in the trend of radon concentration (i.e. rise) that seems to be the actual important one for the present case study, but our next target is to include negative variations too.

COMMENT 5: Western side of study area has recorded more number of seismic events and of higher magnitude as compared to eastern side. Explain reasons?

REPLY 5: The regional seismotectonic framework of the Pollino area lies at the northern edge of the Calabrian subduction zone and represents quite a complex sector of remarkable geodynamic interest still not fully understood, as many studies showing different conclusions demonstrate (Bonini et al., 2011; Frepoli et al., 2011; Spina et al., 2011; Neri et al., 2012). The hypocenters relocalization of the 2010–2013 swarm that has revealed two main clusters, has been subject of a recent work (Totaro et al., 2015), but no indication came about the different number and magnitude distribution of seismic events recorded in the Western side of the area. The only reasonable explanation could lie in the Gutenberg-Richter law: two different clusters of values for magnitude and total number of localized earthquakes seem to suggest that two distinct structures of different dimensions have been activated.

COMMENT 6: Whether the meteorological observatory is located on site, or far from the radon monitoring site, if so how far from the site? Please clarify the situation for both monitors.

REPLY 6: The two radon stations MMN and MMNG installed in the Pollino area acquire simultaneously radon concentration data and local temperature values by means of a specific sensor co-located with the radon one. As explained in the manuscript, all other meteorological parameters daily values (external temperature, pressure, precipitation) employed in this work are obtained as short term (12-24h) weather forecast by an Italian weather forecasting site (http://www.ilmeteo.it/).

COMMENT 7: Authors are requested to explain reasons for the following: a) During summer season, Radon concentration is more pronounced. Why? b) Radon concentration decreases with increase in rainfall. Why? c) Can you elaborate further on your claim with explanation that 'the discontinuity in the radon increase is likely to be associated with a major rainfall that occurred just after the 5.2 magnitude earthquake'?

REPLY 7: a) The reason of this behaviour of radon concentration during summer is likely to be complex and not related to a single cause as different results from previous analyses carried on in different setting indicate. It is likely that precipitation could also play a role, being summer rain much lighter than winter one could be ascribable to minor precipitation events, considering that rain clearly inhibits radon emanation forming a sort of seal into the first layers of soil (see Figures 2 and 3). Incidentally also Oh, Y. and Kim, G. (2015) have recently observed that radon activity was on average higher during the summer than during the winter. b) The answer to the second question raised is closely connected with the point above and actually it has been subject to particular attention in the manuscript. c) The sense of this claim is that we believe that the severe rain event occurred just after the 5.2 magnitude earthquake has influenced a radon increase having a quasi-linear trend starting from the beginning of October (therefore almost a month before the mainshock). In Fig. 5c, where raw radon concentration (yellow dots) is reported together with daily averaged rainfall it is clear this trend from September to the end of November 2012 with the only exception, exactly, of the week after the 35 mm $H_2O$ peak of rain.

COMMENT 8: It is very interesting observation recorded by authors that: For the seismic event of magnitude 4.3, radon concentration gets restored to normal value after 7 days of the event. For the seismic event of magnitude 5.2, radon concentrations continue to increase for more than 30 days after the event. Can you explain the possible reasons?

REPLY 8 The actual physical mechanisms leading to radon emanation anomalies during earthquakes preparation are far from well assessed yet but they are likely to be

connected to the dynamics of fluids taking place in rocks undergoing rapid and dramatic changes of their internal state. While not quantitatively constrained yet, nevertheless it is reasonable to assume that the time needed to recover from this perturbed state (and thus to return to background radon level) is proportional to the overall energy involved in the seismogenic processes and hence to the magnitude of the impending earthquake.

COMMENT 9: Typo-graphical suggestions

REPLY 9: We corrected as suggested.

Oh, Y. and Kim, G. A radon-thoron isotope pair as a reliable earthquake precursor. Sci. Rep. 5, 13084; doi: 10.1038/srep13084 (2015).

Please also note the supplement to this comment:
http://www.solid-earth-discuss.net/se-2016-72/se-2016-72-AC1-supplement.pdf

———————————————————

[Figure]

**Figure1R**: MMN radon concentration (yellow dots) in (Bq/m³)/115 min (a), daily average (b), 14-days moving average (c) from January 2012 to October 2014. Red stars represent the occurrences of the main earthquakes of the sequence. Blue solid and dashed lines represent mean value and 2SD, respectively. a), b) and c) show how the "anomaly" concept may depend from different sampling times of moving average employed.

**Fig. 1.** See supplement for original size

---

## Author Comment (AC2) · 5 Jul 2016

P. Einarsson (Referee)

COMMENT 1: More technical information is needed. I cannot find any information about the sampling technique. There is no mention of what the carrier fluid is, atmospheric air, ground water, geothermal fluid, geothermal gas? What is the depth of sampling, surface, soil, drill hole? This is particularly important in the light of the strong correlation of radon concentration with meteorological factors, (see e.g. page 6, lines 5-6). This correlation is understandable if the sampling takes place at the surface. It may be possible to reduce the disturbing influence of weather by taking samples from deep drill holes (Hauksson and Goddard, 1981, Einarsson et al., 2008). (I note the

words "soil radon emanation" appear on page 8, line 33, and page 9, line 19-20 and 29).

REPLY 1: Radon monitoring stations MMN and MMNG have been both developed with a highly sensitive and efficient active detector to monitor radon concentration in ambient air. The system employs an alpha particles scintillator made by a Lucas Cell integrating front-end electronics, and measures radon concentration with an adjustable acquisition time (default is 115 minutes); its efficiency is 0.06 [CPM /(Bq/m3)] and the minimum detectable concentration is as low as 3.4 Bq/m3 (see also Table 1 in Supplementary Information). The radon detectors of both stations are located in a small room of an isolated building, not disturbed by anthropogenic influences and without any kind of opening and/or aeration system. We have experimentally verified that, for our purposes, the results obtained by means of this setup are equivalent with respect to a radon probe inserted directly in the soil (see Appendix A of Piersanti at al., 2015). We added in the reviewed manuscript (Supplementary Information section) a more exhaustive description of the sampling technique, as suggested.

COMMENT 3: Typo-graphical suggestions or specific comments

REPLY 3: We corrected as suggested.